# Vaccine Hesitancy among Parents and Its Association with the Uptake of Measles Vaccine in Urban Settings in Khartoum State, Sudan

**DOI:** 10.3390/vaccines10020205

**Published:** 2022-01-28

**Authors:** Majdi M. Sabahelzain, Mohamed Moukhyer, Bart van den Borne, Hans Bosma

**Affiliations:** 1Department of Public Health, School of Health Sciences, Ahfad University for Women, Omdurman P.O. Box 167, Sudan; 2Department of Health Promotion, Care and Public Health Research Institute (CAPHRI), Maastricht University, P.O. Box 616, 6200 MD Maastricht, The Netherlands; b.vdborne@maastrichtuniversity.nl; 3Department of Research and Development, Faculty of Applied Medical Sciences, Jazan University, Jizan 45142, Saudi Arabia; moukhyer@hotmail.com; 4Department of Emergency Medical Services, Faculty of Applied Medical Sciences, Jazan University, Jizan 45142, Saudi Arabia; 5Public Health Programmes, School of Medicine, University of Limerick, V94 PX58 Limerick, Ireland; 6Department of Social Medicine, Care and Public Health Research Institute (CAPHRI), Maastricht University, P.O. Box 616, 6200 MD Maastricht, The Netherlands; hans.bosma@maastrichtuniversity.nl

**Keywords:** measles vaccine, vaccine hesitancy, measles vaccine uptake, immunization, Sudan, PACV

## Abstract

Vaccine uptake is one of the indicators that has been used to guide immunization programs. This study aimed to evaluate whether measles vaccine uptake is predicted by measles vaccine hesitancy. A community-based cross-sectional study was conducted in urban districts in Khartoum state in February 2019. Measles vaccine uptake among children was measured as either fully vaccinated or partially/not vaccinated. The Parent Attitudes about Childhood Vaccines (PACV) scale was used to measure measles vaccine hesitancy. Multivariate logistic regression was run to identify the predictors of measles vaccination uptake, controlling for sociodemographic variables, and the adjusted odds ratios (aORs) with 95% CI were calculated. The receiver operator characteristic (ROC) curve was created, and the area under the curve (AUC) for the PACV was computed. Data were collected from 495 participants. We found that measles vaccine hesitancy (PACV scores) predicts the uptake of measles vaccine after controlling for other potential social confounders, such as the mother’s age and the number of children (aOR 1.055; 95% CI 1.028–1.028). Additionally, the ROC for the PACV yielded an area under the curve (AUC 0.686 (95% CI 0.620–0.751; *p* < 0.001)). Our findings show that measles vaccine hesitancy in Sudan directly influences the uptake of the measles vaccine. Addressing the determinants of vaccine hesitancy through communication strategies will improve vaccine uptake.

## 1. Introduction

Measles vaccination prevented about 23 million deaths worldwide between 2000 and 2018. In addition, there has been a 66% decline in measles incidence worldwide and a 73% decline in measles-related deaths during the same period [1]. Nevertheless, the World Health Organization (WHO) reported a 300% increase in measles cases worldwide in the first quarter of 2019 compared to the same period in 2018 [2,3].

It has been reported that measles cases increased by 246% in the African region in 2018 compared to 2016. Five countries accounted for 45% of all globally reported cases of measles, covering the Democratic Republic of Congo, Liberia, Madagascar, Somalia, and Ukraine [1]. In sub-Saharan Africa, the measles case–fatality ratio ranges from 5% to 10%, while in developed countries, fewer than 1 in 1000 children with measles die. In countries with refugee camps and internally displaced people, measles ranks among the top causes of child deaths, with case–fatality ratios between 20% and 30% [4]. Sudan has witnessed measles outbreaks in different parts of the country in the past couple of years, with an increase of 649% in the number of confirmed cases (4978 in 2018) compared to 2017 [3]. Malnutrition and difficulty in getting vaccines to children in Darfur, South Kordofan, and the Blue Nile states worsen the situation for measles in Sudan [5].

Various global efforts, such as the Global Eradication of Measles, WHO Global Vaccine Action Plan 2011–2020 (GVAP), and Immunization Agenda 2030, have identified different strategies for eliminating measles. One of these strategies calls for vaccination of 90% of the target population and 80% of the population in every covered district [1,6,7]. No African country has yet achieved the measles elimination goal [8]. Low vaccination uptake and increasing vaccine hesitancy contribute to the persistent failure to reach approximate herd immunity targets (≥95%) [8,9].

As one of the main indicators used in immunization programs worldwide, vaccine uptake can be predicted by a variety of factors, ranging from individual and interpersonal factors to societal issues such as health and immunization policies [10,11,12]. Research in African countries reported that vaccination rates were lower in rural than in urban areas. Factors such as the child’s age, gender, and birth order and the number of siblings within the household were significantly related to vaccination rates. In addition, parental factors, such as mothers’ age, education, and socioeconomic status, were found to be predictors of vaccine uptake [13,14,15,16,17].

Vaccine hesitancy is well established as one of the most important predictors of vaccine uptake in high-income countries [18,19,20,21]. It has been named in 2019 by the WHO as one of the top 10 global health threats [22]. The Strategic Advisory Group of Experts (SAGE) on vaccine hesitancy has defined vaccine hesitancy as the “delay in acceptance or refusal of vaccines despite availability of vaccination services. Vaccine hesitancy is complex and context-specific, varying across time, place and vaccines.” Vaccine hesitancy is influenced by factors known as the 3Cs model: complacency (low risk perception of vaccine-preventable diseases and the belief that no vaccine is needed), convenience (access issues and constraints), and confidence (the level of trust in a vaccine or provider) [18,19]. Three tools were developed by the SAGE/WHO to measure vaccine hesitancy quantitatively and qualitatively and evaluated in many countries [19,20,23,24,25]. One of these tools, the Vaccine Hesitancy Scale (VHS), which is reliable and has a moderately good convergent validity, was adapted and evaluated in Sudan; however, it has a limitation in predicting the concurrent child’s vaccination status [26].

Vaccine hesitancy and its impact on vaccine uptake and demand are poorly investigated in low-income countries, suggesting a more complex relationship between supply-side and demand-side factors [18,19,20,21,25,26,27,28,29,30]. The Parent Attitudes about Childhood Vaccines (PACV) survey is a widely used tool that was designed to measure and identify vaccine-hesitant parents in different high- and middle-income countries [31,32,33,34,35,36,37]. Moreover, studies in the USA and Tennessee have shown that the PACV survey could predict childhood immunization uptake [38,39,40,41].

In Sudan, the national vaccination coverage is suboptimal for the first and the second dose of the measles-containing vaccine (88% and 72%, respectively) [42]. The reasons behind the low uptake of the measles vaccine are not fully understood. Data from Sudan suggest the existence of measles vaccine hesitancy in Sudan, with several social and behavioral drivers behind this hesitancy [26,29,43]. In this study, we aimed to evaluate whether measles vaccine uptake is predicted by measles vaccine hesitancy. This study is part of a larger community-based, cross-sectional, mixed-methods research project in Khartoum, Sudan, which aimed to inform developing strategies to address the low uptake of measles vaccine in Sudan.

## 2. Materials and Methods

### 2.1. Study Design

The research design was a community-based cross-sectional study and was conducted in two urban districts in Omdurman locality in Khartoum state in February 2019. These two districts were selected for the study because they reflect the typical sociodemographic and socio-cultural situation in Sudan. As the two are in an urban setting, this may ensure a relative exposure to vaccination communication campaigns as well as accessibility to vaccination services. The latter is a prerequisite for the assessment of vaccine hesitancy.

### 2.2. Population and Sampling

#### 2.2.1. Population

The study population included parents/guardians having at least one child aged 2–3 years old. Either mothers or fathers were eligible for participation. If there was more than one child in the same age range in the family, the parents/guardians were asked to answer about only the youngest one, to avoid recall bias. If both mother and father were available, they were asked to nominate one of themselves to complete the questionnaire.

#### 2.2.2. Sampling

This study is part of a large research project on measles vaccine hesitancy in Sudan [10,30]. The sample size was calculated for the whole research using a power analysis for the association between measles vaccine hesitancy and the measles vaccination status (outcome), which showed that at least 386 participants were needed to yield an 80% power to detect an odds ratio of 1.7 at the alpha level (5%). We assumed the prevalence of the outcome, the measles vaccination status among the exposed group (hesitant parents) was 50% [44]. To cover for possible drop-out due to missing information on the important questions during the survey, we recruited more participants to complete a total of 500 participants (parents/caregivers) in the study.

We collected data from parents/caregivers in two different urban districts in Omdurman, Alsharafia (Wad Nubawi’s administrative unit) and Abo Saeed (Abo Saeed’s administrative unit). These two districts have similar characteristics in terms of urbanization, locality (i.e., Omdurman), exposure to vaccination and communication/information campaigns, as well as relative availability of vaccine services as a prerequisite for assessment of vaccine hesitancy. However, people who live in these two districts have different socioeconomic backgrounds (i.e., education, employment, and income levels are higher in Abo Saeed than in Alsharafia). Parents/caregivers were selected in each district using consecutive sampling (convenience sample), as every parent/caregiver meeting the criteria of inclusion (e.g., had a child in the age range) was included in the study until the required sample size was achieved from each district.

### 2.3. Data Collection

Data were collected using a pre-tested, structured questionnaire. Data were collected by eight well-trained graduate female students from the Ahfad University for Women.

#### 2.3.1. Dependent Variable

The dependent variable in this study was measles vaccine uptake by the youngest child, in the age range of 2–3 years (i.e., the measles vaccination status), who was measured as either fully vaccinated (i.e., two doses) or partially (i.e., single dose)/not vaccinated (no dose). First, we asked the parents/guardians to show the vaccination card of their youngest child (2–3 years). If there was no card, then we asked them to report about their child’s measles vaccination status. Only 42.8% showed their children’s vaccination cards; 54.6% reported that they had cards, but they did not show them. We excluded from the analysis all parents/guardians who reported they did not know their children’s vaccination status.

#### 2.3.2. Independent Variables

In this study, we used the Parent Attitudes about Childhood Vaccines (PACV) to measure measles vaccine hesitancy as the main independent variable. The PACV includes 15 items categorized in three domains: immunization behavior (items 1 and 2), perceived safety and efficacy (items 7–10), and general attitudes and trust (items 3–6 and 11–15). The items in this scale were scored using a 5-point Likert scale ranging from strongly agree to strongly disagree. For measuring vaccine hesitancy, we combined the items for perceived safety and efficacy and the general attitude and trust items. Each of the 15 PACV survey items was scored as follows: Hesitant responses were assigned a 2, “don’t know or not sure” a 1, and non-hesitant responses a 0. The two behavior items, i.e., items 1 and 2, (see Appendix A) were scored as 2 for the hesitant response and as 0 for the non-hesitant response, as the “don’t know” responses were excluded as missing data as suggested by Opel et al. [31,38]. The raw total PACV score was calculated by simply summing each item. The total raw score ranged from 0 to 30. Then, the raw score was converted to a 0–100 scale [31,38]. Cronbach’s alpha was computed for this scale (Q3–Q15), which was 0.62 [26].

Additional independent variables in this study were sociodemographic characteristics of the family, which included the mother’s education, which was measured at four levels as described by the ministry of education, none (not attended any formal or non-formal education), primary (lasting 8 years, from Grade 1 to Grade 8), secondary (ages 14 to 16 can attend secondary education, which lasts 3 years), and university level (i.e., diploma, bachelor’s, and postgraduate degrees); the income level of the family, which was self-ranked by the study participants on three levels (high, medium, and low); mothers’ employment; the number of children who were aged <5 years in the family; and the total number of household members.

### 2.4. Statistical Analysis

Data analysis was performed using Statistical Package for Social Sciences (SPSS) (V 24). Frequencies were generated for the sociodemographic characteristics of the family. Frequencies of the PACV items were calculated. The chi-square test and Fisher’s exact test (when the count in the cells is less than 5) were run to identify factors univariate associated with the dependent variable (i.e., measles vaccination status). Additionally, Pearson coefficients were calculated to assess the correlations between measles vaccine uptake and the socioeconomic factors and measles vaccine hesitancy (PACV). For both the chi-square test and Pearson correlations, a *p*-value of less than 0.05 was considered statistically significant. Multiple logistic regression was performed to identify the predictors of measles vaccination status controlling for sociodemographic variables, and the adjusted odds ratios with 95% CI were calculated. Only correlates and factors that were significantly related to the uptake of measles vaccine were included in the multiple regression analysis. The receiver operator characteristic (ROC) curve was executed. In addition, the area under the curve (AUC) for the PACV was computed to evaluate the ability of the PACV to distinguish and predict the child’s measles vaccination status.

## 3. Results

### 3.1. Descriptive Statistics and Associations of the Sociodemographic and Parental Perceptions about the Measles Vaccine with the Uptake of the Measles Vaccine

As shown by Table 1, 495 participants from Omdurman city were included: 30.7% from Wad Nubawi’s district and 69.3% from Abo Saeed district. The majority of the participants were mothers (87.2%), with low participation by fathers (only 4.6%). The mean age of the mothers who participated in the study was 31.14 (SD = 5.73). About half of the participants (50.1%) had completed university, followed by those who attended secondary schools (34.3%). Nearly, three-quarters of the participants (74.7%) were housewives. About 79.0% of the participants self-ranked their income levels as a medium. The majority of the participants mentioned that they have either 1 or 2 children (44% and 45.9%, respectively). About a third of the participants reported that they have 3–4 members in their households.

Moreover, we found that measles vaccine uptake was highly associated with the mother’s employment, as self-employed mothers were more likely to get their children only partially vaccinated or not to get them vaccinated, followed by mothers who were workers and housewives (*p*-value < 0.017). The number of children was associated with measles vaccine uptake, as families with 3 or more children were more likely to get their children only partially vaccinated or not get their children vaccinated with measles vaccine compared to mothers with 1 child (*p*-value = 0.041), see (Table 1). Families with 5 or more members had a lower tendency to get their children fully vaccinated than families with less than 3–4 members.

### 3.2. Pearson Correlations between Measles Vaccine Uptake and the Socioeconomic Factors and Measles Vaccine Hesitancy (PACV)

Pearson correlation was run to assess, firstly, the relation between the uptake of measles vaccine and the socioeconomic factors and measles vaccine hesitancy (PACV) and, secondly to assess multicollinearity between the correlates to avoid its negative effect on the multivariate analysis. Table 2 shows that the uptake of measles vaccine among children is strongly correlated with the PACV scores and the number of household members (r = 0.22 and 0.14; *p*-value < 0.01, respectively), weakly correlated with mother’s employment (r = 0.091; *p*-value < 0.05), and negatively correlated with mother’s education (r = −0.091; *p*-value < 0.05).

### 3.3. Multiple Logistic Regression Analysis

To assess whether measles vaccine hesitancy (PACV scores) predicts the uptake of the measles vaccine, we ran a multivariable logistic regression model adjusting all socio-demographic variables that were significantly associated with the uptake of the measles vaccine at the bivariate level. The logistic regression analysis results are presented in Table 3. We found that measles vaccine hesitancy (PACV scores) predicted the uptake of measles vaccine (b = 0.053; Waldχ2 = 16.812; *p*-value < 0.00; aOR = 1.054 (95% C.I, 1.028–1.081)).

### 3.4. ROC Curve

The PACV’s scores were analyzed using ROC analysis to describe their ability to predict the child’s measles vaccination uptake. The nonparametric analysis of the ROC for the PACV yielded an area under the curve (AUC) of 0.686 (95% CI = 0.620, 0.751) (*p* < 0.001; Figure 1). This reveals that the test can significantly predict measles vaccine uptake among children.

## 4. Discussion

The present study aimed to assess whether measles vaccine uptake can be predicted by measles vaccine hesitancy (PACV scores) in two urban areas in Khartoum state.

Our study’s findings showed that about 12% of the children (2–3 years) were either under-immunized, with only a single dose of measles vaccine, or unimmunized. This finding mirrors results from the annual statistical report in 2019, as the rate of receiving the first dose of measles vaccine in Khartoum state and at the national level was estimated at 88% [42,45]. Our data were collected only from urban areas, which represent about 30% of Sudan [46]. We purposively targeted urban areas to ensure the relative availability of vaccines services and thus control other factors related to vaccine access issues. People in rural areas in Sudan are underserviced with vaccines services, which reflect geographical and socioeconomic inequality. Although measles vaccine coverages in both urban and rural areas are suboptimal at the national level, the number of children in urban areas who received the second dose of measles vaccine is 8% higher than the number of children in rural areas (85.4% and 77.5%, respectively) [46].

Official reports in Sudan indicate that measles is the third cause of death among children under 5 and the first among vaccine-preventable diseases. [45,47]. WHO recommends that countries that aim to eliminate measles should achieve ≥95% coverage with both doses (i.e., the first and the second one) of all children in each district [48]. Studies from different African countries, including Sudan, have suggested that countries with vaccine access issues are far away from achieving the measles elimination goal [13,14,15,16,17,29].

Importantly, our study underscored that measles vaccine hesitancy among parents influences the uptake of measles vaccine among children. Additionally, the nonparametric analysis of the ROC for the PACV yielded an area under the curve (AUC) of 0.686 (95% CI 0.620–0.751) (*p* < 0.001). Both findings reveal that the PACV can significantly predict measles vaccine uptake among children. As the present study was conducted in a low-income context, it supports the predictive validity of the PACV in determining the vaccination status of children as shown in previous studies in high and middle-income countries [38,39,40,41].

Given the complexity of the relationship between supply-side and demand-side factors and measles vaccine hesitancy as suggested by previous studies in Sudan [29,43], different approaches should be adopted to address vaccine hesitancy to increase the uptake of measles vaccine in Sudan. For the demand side, previous findings in Sudan have shown that parental exposure to anti-vaccine information is predicting measles vaccine hesitancy among them [43]. We suggest that communication strategies be theory and evidence based to address the behavioral determinants of vaccine hesitancy, such as knowledge of, beliefs about, and attitude toward measles and measles vaccine. In a rapidly evolving digital world, there is a need for an environment that provides the public and health care providers with health-related information about vaccines as well as addresses infodemics. Therefore, communication at the community level should be enhanced and sustained by using not only mass media (TV, radio, preaching in mosques, etc.) but also social media (websites, Facebook, Twitter, etc.).

For the supply side, one major access-related issue might be the presentation of the measles vaccine in multiple doses (10 doses per vial). This results in 1–2 sessions being conducted per week in many low-income countries to comply with the open-vial policy, which recommends discarding the 10-dose vial 6 h after opening the vial if unused as well as to reduce vaccine wastage. Furthermore, these access issues may negatively affect vaccine acceptance, as parents actively trying to get their child vaccinated with the measles vaccine turn away when the provider refuses to open the measles vaccine vial [13,14,15,16,17,28,29]. Evidence from some African countries suggests that changing the number of doses per vials from 10 to 5 would double the number of opportunities to open a vial compared to its presentation in 10-dose vials [49,50,51,52]. We suggest that an analysis of the immunization policy as well as cost-effective analysis is needed to anticipate what will happen if the 10-dose vial of measles vaccine (i.e., open-vial policy) is shifted to a 5-dose vial in Sudan as it has succeeded in many LMIC countries [49,50,51].

Interestingly, although the mother’s education and the income level of the family were not significantly associated with the uptake of the measles vaccine, our findings showed that children of mothers who are not educated are twice as likely to be only partially vaccinated or unvaccinated than children of mothers who had attained university education (21.4% and 10.1%, respectively). In addition, the proportion of partially vaccinated/unvaccinated children in families who reported a low income level is more than twice than those in families who reported a high income level (17.6% and 7.1%, respectively). These findings may underscore socioeconomic inequalities related to the uptake of the measles vaccine. In terms of intervention strategies, these lower socioeconomic groups should be prioritized as early target groups.

## 5. Limitations

We acknowledge some limitations related to our study. Therefore, the study’s findings should be interpreted within the context of this study. One limitation was that our study was conducted in two urban districts in Omdurman locality in Khartoum state, which may have led to a selection of relatively higher-educated families, as about half of the female participants (50.1%) had attained university education. This rate is higher than the average rates for the females who are attending university (about 15% and 30% at the national and Khartoum-state levels, respectively) [52]. These districts also have a higher rate of vaccination (88%), as has been shown by our findings, which can be explained also by the level of education, though it was not statistically significant, as well as the relative availability and accessibility of vaccination services. However, less than half of the participants (42.8%) showed their children’s vaccination cards, which may underestimate the non-vaccination rate. From a gender perspective, we missed fathers’ perceptions and perspectives in this study, as data were collected from 10:00 am to 5:00 pm, the working hours of most fathers. However, Sudanese mothers, as in many other African countries, are mostly the first persons responsible for the health and prevention of sickness of their children/family and should know the health situation best [53]; therefore, some parents preferred the mother’s participation in the study to the father’s participation. Given the unavailability of the data, we did not control for all possible confounders.

## 6. Conclusions

Our study findings underscored that measles vaccine hesitancy influences the uptake of measles vaccine and that the PACV scores predict the immunization status of Sudanese children. In light of these findings, we suggest that intervening on measles vaccine hesitancy will have a direct impact on the uptake of the measles vaccine in Sudan. Improving the vaccination status of Sudanese children could be achieved by developing and implementing immunization communication strategies that address the determinants of vaccine hesitancy, which should increase the confidence in the measles vaccine by correcting misinformation, debunking myths and rumors about vaccines, and scientifically addressing the vaccine safety issues. Intervention strategies should prioritize parents in lower socioeconomic groups as they showed lower uptake of measles vaccine.

## Figures and Tables

**Figure 1 vaccines-10-00205-f001:**
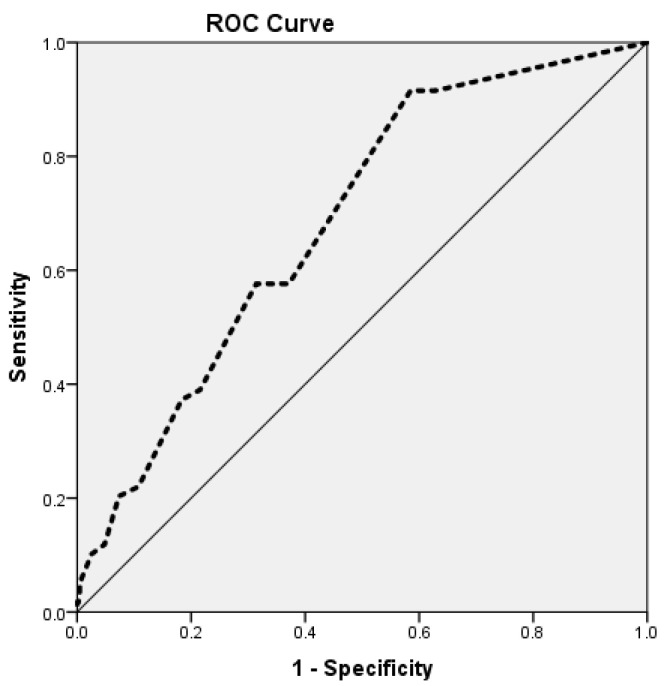
Receiver operating characteristic (ROC) analysis of the PACV scores for the screening of vaccine hesitancy.

**Table 1 vaccines-10-00205-t001:** Association of the characteristics of the surveyed parents/guardians with measles vaccination status. *n* = 495.

Characteristics		Measles Vaccination Uptake/Status					
		Total	Fully Vaccinated		Partially/Unvaccinated		*p*-Value
		*n* = 495 (%)	*n* = 436	%	*n* = 59	%	
Area of the study	Alsharafia	152 (30.7)	128	84.2%	24	15.8%	0.077
	Abo Saeed	343 (69.3)	308	89.8%	35	10.2%	
Mother’s Education	Illiterate	14 (2.8)	11	78.6%	3	21.4%	0.162
	Primary	63 (12.7)	51	81.0%	12	19.0%	
	Secondary	170 (34.3)	151	88.8%	19	11.2%	
	University	248 (50.1)	223	89.9%	25	10.1%	
Mother’s Employment	Housewife	370 (74.7)	323	87.3%	47	12.7%	0.017 *^,b^
	Student	11 (2.2)	10	90.9%	1	9.1%	
	Worker	14 (2.8)	12	85.7%	2	14.3%	
	Officer	50 (10.1)	48	96.0%	2	4.0%	
	Professional (e.g., Engineer)	33 (6.7)	32	97.0%	1	3.0%	
	Self-employed	16 (3.2)	10	62.5%	6	37.5%	
	Others	1 (0.2)	1	100.0%	0	0.0%	
Income Level	High	70 (14.1)	65	92.9%	5	7.1%	0.268
(Self-Ranking)	Medium	391 (79.0)	343	87.7%	48	12.3%	
	Low	34 (6.9)	28	82.4%	6	17.6%	
Number of Children	1	218 (44.0)	185	84.9%	33	15.1%	0.041 *
	2	227 (45.9)	209	92.1%	18	7.9%	
	3 and more	50 (10.1)	42	84.0%	8	16.0%	
Total number of household’s members	3–4	178 (36.0)	169	94.9%	9	5.1%	0.002 *
	5–6	159 (32.1)	134	84.3%	25	15.7%	
	7 and more	158 (31.9)	133	84.2%	25	15.8%	

* Statistically significant, ^b^ = Fisher’s exact test.

**Table 2 vaccines-10-00205-t002:** Pearson correlations between measles vaccine uptake and the socioeconomic factors and measles vaccine hesitancy (PACV).

Socioeconomic Factors	Area of Study	Mothers’ Age	Mothers’ Education	Mother’s Employment	Family Income Level	Number of Children	Number of Household Members	PACV Scores	Measles Vaccine Uptake
Area of Study	X								
Mothers’ Age	0.116 *	X							
Mothers’ Education	0.160 **	0.006	X						
Mother’s Employment	0.034	0.045	0.191 **	X					
Family Income Level	−0.175 **	0.005	−0.293 **	−0.183 **	X				
Number of Children	−0.017	−0.013	−0.053	−0.038	0.067	X			
Number of Household Members	0.047	0.402 **	−0.266 **	−0.067	0.144 **	0.219 **	X		
PACV Scores	−0.039	0.037	−0.011	0.031	0.009	0.103 *	0.082	X	
Measles Vaccine Uptake	−0.080	0.091 *	−0.091 *	−0.014	0.073	−0.048	0.139 **	0.222 **	X

* Significant at the 0.05 level (2-tailed); ** Significant at the 0.01 level (2-tailed), *n* = 495.

**Table 3 vaccines-10-00205-t003:** Predictors of partial vaccination or no vaccination with measles vaccine in Khartoum state, Sudan.

Predictors	OR (95% CI of OR)	aOR (95% CI of OR)
PACV scores	1.053 * (1.030–1.078)	1.054 * (1.028–1.081)
Age of mother	1.049 * (1.001–1.100)	1.020 (0.966–1.076)
Number of household’s members **		
3–4 (ref)		
5–6	3.503 * (1.582–7.757)	3.317 * (1.450–7.589)
7 and more	3.530 * (1.594–7.817)	2.528 * (1.044–7.881)
Mother’s employment		
Housewife (ref)		
Student	0.687 (0.086–5.491)	0.575 (0.065–5.064)
Worker	1.145 (0.249–5.279)	0.922 (0.185–4.586)
Officer	0.286 (0.067–1.217)0	0.317 (0.073–1.377)
Professional (e.g., engineer, doctor)	0.215 (0.029–1.609)0	0.231(0.030–1.770)
Self-employed	4.123 * (1.432–11.870)	3.189 (0.868–11.718)

* *p* < 0.05, aOR = adjusted odds ratio; ref = reference category; ** the number of household members was strongly related to the mother’s education and the number of children (r = −0.266 and 0.219, respectively; *p* < 0.01), therefore, only the number of household members was included in the multiple regression analysis.

## Data Availability

Not applicable.

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
