# Peer review of "Vaccine Hesitancy among Parents and Its Association with the Uptake of Measles Vaccine in Urban Settings in Khartoum State, Sudan"

_vaccines, 2022, doi:10.3390/vaccines10020205_

Round 1

Reviewer 1 Report

This article analysed the effect of vaccine hesitancy on uptake of measles vaccines in Sudan. The topic is of significant relevance to the design and implementation of public health policies meant to address recurrent health problems among children in Sudan. Although the authors have attempted to present good article, there are some concerns that may jeopardize the acceptability of this paper in the current form. These major issues include:

  • The introduction is not well articulated. It would be proper to start with some background information on incidences of measles among children in developing countries and Sudan in particular. Then tell us efforts already made to address the problem with emphasis on vaccination. Then the issue of vaccine hesitancy and its association with uptake of measle vaccines can be discussed along with other associated variables. The author needs to also do a lot of work in setting this study in the context of what had been known. It is presently not clear what new knowledge this study is generating given the volume of existing body of knowledge on the subject matter.
  • The study was conducted in two urban centers in Omdurmen. It is then not clear how this study is being representative of Sudan as a whole. Instead, one would expect the author to relay the title to the district where the data were collected.
  • Although classifying the children into two groups of fully vaccinated and not fully vaccinated/not vaccinated meet the requirements of using logistic regression, it is not clear why distinctions were not made between those that were not fully vaccinated and those that were not vaccinated at all. This may then compel a change in the estimated model.
  • The author did not explain the specification of the logistic regression model under the methodology section. It is also not clear how the independent variables like mothers’ income were measured. Given that the endogeneity tendency of vaccine hesitancy, it is not clear why the authors assumed that this variable to be exogenous. A more rigorous econometric model that takes cognizance of the potential endogeneity concern should have been used.
  • The compilation procedures of the vaccine hesitancy that made the value range between 0 and 30 needs to be clarified.

Author Response

Response to Reviewer 1 Comments

  • The introduction is not well articulated. It would be proper to start with some background information on incidences of measles among children in developing countries and Sudan in particular. Then tell us efforts already made to address the problem with emphasis on vaccination. Then the issue of vaccine hesitancy and its association with uptake of measle vaccines can be discussed along with other associated variables. The author needs to also do a lot of work in setting this study in the context of what had been known. It is presently not clear what new knowledge this study is generating given the volume of existing body of knowledge on the subject matter.

Response: The introduction is now articulated using your suggestions including the sequence of the topics, adding the efforts that were made to address measles elimination worldwide as well as contextualizing what is known and unknown within the context of Sudan.

  • The study was conducted in two urban centers in Omdurmen. It is then not clear how this study is being representative of Sudan as a whole. Instead, one would expect the author to relay the title to the district where the data were collected.

Response: We amended the title of the study to reflect the urban area in Khartoum state, where the data was collected: “Vaccine hesitancy among parents and its association with the uptake of measles vaccine in urban settings in Khartoum state, Sudan”

  • Although classifying the children into two groups of fully vaccinated and not fully vaccinated/not vaccinated meet the requirements of using logistic regression, it is not clear why distinctions were not made between those that were not fully vaccinated and those that were not vaccinated at all. This may then compel a change in the estimated model.

Response: We put the partially vaccinated children and not vaccinated children in one groups for two reasons: Firstly, we used the definition of vaccine hesitancy by the WHO/SAGE which define vaccine hesitant people as those who delay vaccine acceptance or refusal. Accordingly, we added the respondents who did not vaccinate their youngest child to those who vaccinate with only one dose (in one group forming about 12% of the all participants). Secondly, the number of not vaccinated at all was very small, therefore, we added them to the partially vaccinated. As the underestimation of non-vaccination (about 2.8%) may have been influenced by the recall and response bias of the participants, as sometimes we relied on parental recall when there is no vaccination card to assess child’s vaccination status. We already reported this valid point in the methods section (2.3.1. Measurement) that “Only 42.8% showed their children vaccination cards, however, 54.6% reported that they had cards but did not show them”.

  • The author did not explain the specification of the logistic regression model under the methodology section. It is also not clear how the independent variables like mothers’ income were measured. Given that the endogeneity tendency of vaccine hesitancy, it is not clear why the authors assumed that this variable to be exogenous. A more rigorous econometric model that takes cognizance of the potential endogeneity concern should have been used.

Response: We amended the statistical analysis section to clearly explain the specification of the logistic regression model #Lines 179-184 “For both, Chi-square test and Pearson correlations, a p-value of less than 0.05 was considered statistically significant. Multiple logistic regression was performed to identify the predictors of measles vaccination status controlling for sociodemographic variables and the adjusted odds ratios with 95% CI were calculated. Only correlates and factors that were significantly related to uptake of measles vaccine were included in the multiple regression analysis.”

We added a description on how we measured the sociodemographic characteristics including the income level of the family, #Lines 162-170: “Additional independent variables in this study were sociodemographic characteristics of the family that included the mother’s education, which was measured at four levels as described by the ministry of education, including none (not attended any formal or non-formal education), primary (this level lasts eight years from Grade 1 to Grade 8), secondary (ages 14 to 16 can attend secondary education, which lasts three years) and university level (i.e. diploma, bachelor’s, and postgraduate degrees); income level of the family, which was self-ranked by the study participants on three levels (i.e. high, medium, and low); mothers’ employment; the number of children who were aged <5 years in the family; and the total number of household’s members.”

We already addressed carefully the potential confounders by including not only the statistically significant factors but also the correlates in the multiple regression analysis. However, we might be did not control for all possible confounders. Given the unfeasibility of the methods (instrumental variables/packages) that the reviewer advises as well as unavailability of the data, we think that we controlled for the most important confounders. Nevertheless, we added that we did not control for all possible confounders in the limitation section: “Given the unavailability of the data, we did not control for all possible confounders”

  • The compilation procedures of the vaccine hesitancy that made the value range between 0 and 30 needs to be clarified.

Response: We clarified the compilation procedures of PACV scores in #Page 4, #Lines 154-161:

“Each of the 15 PACV survey items was scored as; Hesitant responses are assigned a 2, ‘don’t know or not sure a 1, and non-hesitant responses a 0. The two behavior items (i.e. items 1 and 2), [see Table A1], are scored as 2 for the hesitant response and 0 for the non-hesitant response, as the “don’t know” responses were excluded as missing data as suggested by Opel et al. [31,38]. The raw total PACV score was calculated by simply summing each item. The total raw score ranged from 0 – 30. Then, the raw score was converted to a 0 – 100 scale [31,38]”

Reviewer 2 Report

When reporting the logistic regression results I would have liked the b and ê­“2Wald values to have been reported. In the text they should describe the logistic regression results as this example I leave here (b idade = 0.026; ê­“2Wald(1) = 0.110; p = 0.740), instead of simply putting the confidence interval values.
When reporting the results of Pearson correlations, instead of putting for example (-0.091; p-value < 0.05), it would be more correct to put (r = -0.091; p < 0.05). 

Author Response

Response to Reviewer 2 Comments

Comments and Suggestions for Authors

When reporting the logistic regression results I would have liked the b and ê­“2Wald values to have been reported. In the text they should describe the logistic regression results as this example I leave here (b idade = 0.026; ê­“2Wald(1) = 0.110; p = 0.740), instead of simply putting the confidence interval values.
When reporting the results of Pearson correlations, instead of putting for example (-0.091; p-value < 0.05), it would be more correct to put (r = -0.091; p < 0.05). 

Response: Now, we have reported the values of the correlation and the logistic regression as you suggested #Line 2016-218 and #Line 229.

Reviewer 3 Report

The authors conducted a study to confirm vaccine hesitancy was related to a low rate of vaccine uptake, the conclusion of which could be easily predicted and no surprise. Even after reading the whole manuscript, the findings of the study did not seem to give any specific solution for that. The final paragraph of Limitation is confusing since the authors’ opinion was not validated by the statistical analyses in this study. Considering these, the current version of the manuscript would not take broad interest from readers nor be a valuable addition to the existing knowledge on the current situation of measles vaccination.

Author Response

Response to Reviewer 3 Comments

The authors conducted a study to confirm vaccine hesitancy was related to a low rate of vaccine uptake, the conclusion of which could be easily predicted and no surprise. Even after reading the whole manuscript, the findings of the study did not seem to give any specific solution for that. The final paragraph of Limitation is confusing since the authors’ opinion was not validated by the statistical analyses in this study. Considering these, the current version of the manuscript would not take broad interest from readers nor be a valuable addition to the existing knowledge on the current situation of measles vaccination.

Response: Although the comments of the reviewer are more general, we added two paragraphs in the Discussion section to recommend different approaches/solutions to address measles vaccine hesitancy and the low uptake of measles vaccine through communication strategies and policy development (changing current measles vaccine policy) as suggested by the reviewer. Additionally, we revised the limitation paragraph, and we added some statistics and references to support our opinions. Furthermore, we used the comments and the suggestions of the other reviewers to improve the manuscript in particular the introduction and the methods sections.

Round 2

Reviewer 3 Report

(There are no comments)